# Spatial Patterns, Possible Sources, and Risks Assessment of Soil Potentially Toxic Elements in an Open Pit Coal Mining Area in a Typical Arid Region

**Abdugheni Abliz** [1,*,†], **Bilal Imin** [2,†] **and Halidan Asaiduli** [1]

1    College of Geography and Remote Sensing Sciences, Xinjiang University, Urumqi 830046, China
2    College of Biology and Geography, Yili Normal University, Yining 835000, China
*    Correspondence: abdugheni.abliz@xju.edu.cn
†    These authors contributed equally to this work.

**Abstract:** Intensive mining activities in large-scale opencast coal mines have had a significant impact on the local environment. Elements that are potentially harmful to the environment are brought to the surface from deep underground, altering the geochemical conditions for their transport and redistributing them to the surface, causing serious local pollution. However, in-depth studies of toxic metal contamination in soils of arid coal mining areas have not yet received the attention they deserve. Although previous studies have conducted a great deal of research on heavy metal elements in surface coal mine soils, there are few studies related to the more seriously polluted surface coal mines in the arid regions of Northwest China, and there are no in-depth studies on the ecology of soil heavy metal contamination, health risks and source analysis according to the authors' knowledge. To make up for this shortcoming, the present study takes Zn, Cu, Cr, Pb, Hg, and As in the surroundings of the tertiary coal mines in the Hongsachuan Mining Area (northern Xinjiang, China) as an example. The health, ecological risks, and pollution sources of heavy metal elements in surface coal mine soils were comprehensively analyzed. The results showed that the average concentrations of Cr, Hg, and As in the soils of open-pit coal mines greatly exceeded the corresponding provincial background values, with the Cr content exceeding China's soil environmental quality standard I and the As content even higher than standard II (GB15618-1995). Geostatistical and multivariate statistical results showed that the six metals analyzed in this study can be divided into four groups, as follows. Group 1 included Zn and Cu and was mainly controlled by natural sources related to soil parent materials. Group 2 consisted of Cr and Hg and was associated with industrial practices. Group 3 was explained by As and was mainly from coal combustion during the mining activities. Group 4 was Pb and was dominantly from natural sources, together with vehicular emission during the mining activities, indicating a mixed source. Potential ecological risk index (PER) values exhibited low ecological risk in contaminated soils with Zn, Cu, Pb, and Cr, and only 10% of As samples exhibited moderate risks, while 77% of Hg samples posed ecological risks at different level, implying that Hg was the main contributor for comprehensive risk index (RI). Regardless of non-carcinogenic and carcinogenic health risk assessment, As was the primary risk element followed by Cr, and children tended to have a higher health risk than adults. In this paper, statistical methods, pollution assessment methods, and potential ecological risk models are skillfully combined, and relevant conclusions are drawn based on the human and economic geographical background information of the study area. The results can provide references for the investigation and evaluation of soil heavy metals and quantitative analysis of pollution sources in the same type of areas. In order to grasp the pollution level of potential toxic elements in the soil of large open-pit coal mines in arid areas, effective source-cutting measures are taken to provide data support the sustainable management of coal mines and local soil safety utilization measures.

**Keywords:** arid mining area; spatial analysis; multivariate statistical analysis; risk assessment

## 1. Introduction

During mining and smelting, a wide range of potentially environmentally hazardous heavy metals are exposed at the surface and enter the surrounding soil through tailings, effluent discharges, tailings piles, atmospheric deposition, and surface runoff [1]. The existence of potentially toxic elements (PTEs) in the environment poses an adverse effect on human health and a long-term serious risk on environment [2]. The presence of these PTEs is one of the most severe pollutants in terrestrial ecosystems due to their toxicity, biological accumulation, and their potential threat to human health and ecosystems [3,4]. In recent decades, soil PTEs pollution has become a globally recognized environmental issue and has received increasing attention. A wide range of soil PTEs pollution events have been found in agricultural, aquifer, urban, and industrial areas which were induced by natural sources mainly related to geological parent materials and anthropogenic sources, including agriculture, metalliferous mining, industry, waste materials, and atmospheric deposition [5,6]. It is well documented that soils around coal mining areas are considered severely polluted zones, among areas threatened by heavy metal pollution, due to the impact of frequent and sustained mining activities, which are one of the major sources of toxic PTEs in the environment [7].

Technological progress in the mining industry has led to a significant growth in mineral production and contributes to the economic growth of different countries [8]. As the world's largest producer and consumer of coal, it is estimated that coal takes 50–70% of the whole country's energy consumption [9]. Along with the shifts of the coal resources developing center from middle to the northwest, a large-scale, high-intensity coal mining activity has been threatening the original fragile eco-environment a further step [10]. Moreover, soil pollution induced by PTEs has become a serious problem due to barely-decomposed soil microbes that are transformed into more risky compounds when they exceed certain thresholds [11]. Therefore, it is crucial to identify heavy metal sources before taking any specific measures to remediate soil quality. Multivariate statistical analyses and other statistical approaches have been widely applied to identify the coming sources of PTEs in soils [12]. GIS-based geostatistical interpolation techniques are used to evaluate and predict the hazard levels and spatial distribution variability of soil heavy metals. Previous studies proved that multivariate statistical analysis combined with geostatistical technics is an effective method for reflecting the relationships, spatial patterns, and influencing factors of soil parameters, including PTEs [13]. Thus, this research applied multivariate and geostatistical methods to identify the sources of six metals in the soils of the mining area in Junggar basin.

In arid ecosystems, intensified anthropogenic activities could easily lead to ecological degradation as the area is rather fragile, and restoration would be difficult if it was contaminated [14]. Industrial activities such as mining and smelting in arid regions often release into the environment, which in most cases brings the serious pollution of PTEs [15]. Thus, ecosystems have been severely influenced and polluted by the PTEs such as Cd, As, Zn, Cr, and Pb, which are often toxic and risky [16]. Soil PTEs contents, which exceeded the maximum allowable limit criteria in the buffer zones of vehicle, mining, and industrial enterprises, bring severe potential hazards not only to the eco-environment but also to human health [17]. Harmful amounts of PTEs can enter the human body from contaminated soil via three exposure pathways—direct or indirect ingestion, inhalation, and dermal contact—and organisms have the ability to accumulate theses metals through air, food, and water [18–20]. It is well known that excessive accumulation of PTEs in the human body may cause several disorders, including developmental delays, behavioral abnormalities, cardiovascular, cancer damage of kidneys, and malfunctioning of the nervous system [21–24]. Therefore, it is crucial to evaluate both ecological and human health risks caused by the contaminants in highly industrialized areas [25].

Abundant coal resources, convenient transportation, and huge market conditions of Hongshaquan open pit coal mining area have been attracting more industrial enterprises, making it one of the three largest coal mining areas in the Junggar basin. The main

industrial activities in the area are coal-power plants, new material, and mining and chemical industries. The north part of it is the Kalamaili national nature reserve, south is the residential area, the climate condition is extreme, and the ecosystem is weak and fragile. If polluted, it would cause a severe threat to the eco-environment and human health of those who living in the surroundings. Therefore, it is significant to identify the possible coming sources of PTEs and to assess the risks both on ecology and human health for the sustained and stable development of reginal ecology, economy, and society. The specific objectives of this research were: (1) to estimate the pollution characteristics of PTEs in soils of coal mining area; (2) to characterize the heavy metal concentrations and possible coming sources by multivariate statistical analysis together with geostatistical analysis; (3) to evaluate the risks of metals for ecology and human health based on assessment models.

## 2. Materials and Methods

### 2.1. Study Area

The Hongshaquan open pit coal mining area is located east of the Zhundong Economic and Technological Development Park, and east of the Junggar basin. The terrain is high in the southeast and low in the northwest (Figure 1). The area is a typical continental arid desert climate with an average annual temperature of 7 °C and precipitation of approximately 106 mm, while annual evaporation is 1202–2382 mm. It is windy all year round, usually has northwesterly wind with levels of 4–5, and the maximum wind reaches level 10 in spring, always accompanied by sandstorms. The area is mainly sandy-bare land with scarce vegetation, and the ecological condition is extremely sensitive and fragile. The soil is mainly desert saline-alkali soil, and the soil is silty sand. Mainly through a provincial highway to connect to the outside world, coal slag on both sides of the road is the main source of pollution [26].

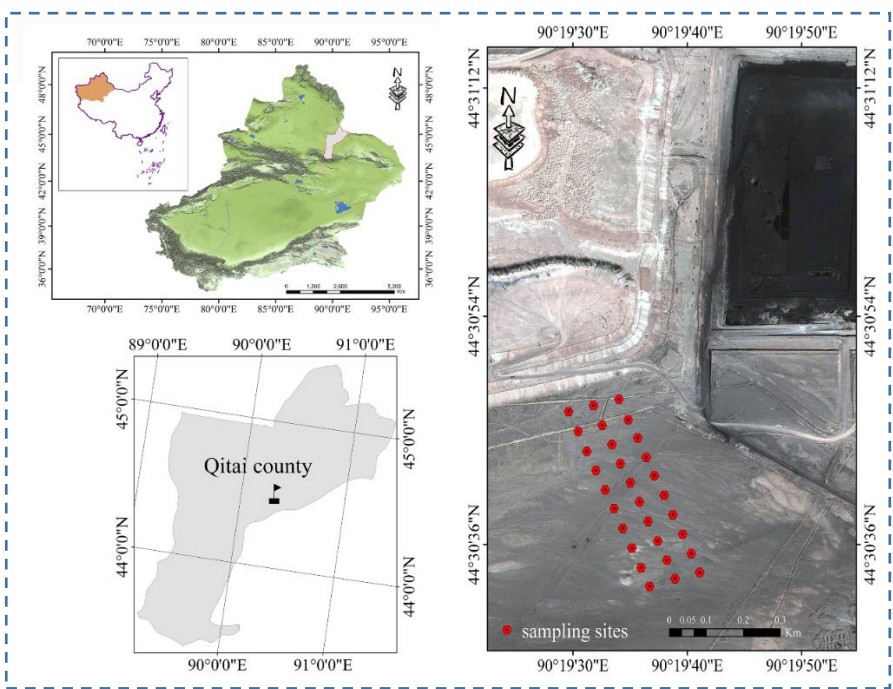

**Figure 1.** Location maps of the study area and distribution of sampling sites.

### 2.2. Sample Collection and Chemical Analysis

For identifying the impacts of PTEs on soils around the open pit coal mine, three sampling lines were selected according to perennial wind direction, 30 soil samples were ground with a wooden stick, and from the 0–20 cm layer the horizontal average distance interval was taken of approximately 50 m. In order to pinpoint the exact positions of all sampling points, the global positioning system (GPS) was applied to record

the exact coordinates. All soil samples were air-dried at room temperature (20–24 °C) and sifted through a 2 mm nylon mesh sieve to remove stones and other debris. About 0.5 g of soil samples were weighed and placed in polyethylene bags to analyze the heavy metal contents of Zn, Cu, Pb, Cr, Hg, and As. The Hg and As values were analyzed by the dual-channel automatic atomic fluorescence spectrometer (PF6–2), and the detection limits of these metals were lower than 0.001 and 0.01 µg·kg$^{-1}$ with the degree of precision $\leq$ 1.0%. In this study, the Hitachi Z-2000 atomic absorption Spectro-photometer was used to determine the Zn, Cu, Cr, and Pb concentrations under the detection limits of 0.001, 0.01, 0.004, and 0.002 mg·kg$^{-1}$. The quality control (QC) was completed using GSS (GBW07401, GSS-8) [27] standard soil sample, provided by the national research center for geo-analysis of China, and the recoveries were within the allowable range for all the metals in the soil [28]. Samples were first digested by the triacid dissolution method (HNO$_3$-HF-HClO$_4$), then determined for As, Zn, Cr, Cu, and Pb using a SOLLARAAM6 atomic absorption spectrometer from Thermos Fisher Scientific, Waltham, MA, USA. To ensure data accuracy, experimental measurements were quality-controlled with 20% parallel samples, blanks, and GSS-1 (GBW07401) [29], a standard for soil composition analysis, with a 5% margin of error. All vessels used in the experiments were soaked in a 10% HNO$_3$ solution for over 24 h. Experimental reagents were of the highest purity, and the water used for the experiments was deionized water.

### 2.3. Statistical and Geo-Statistical Analysis

Pearson's correlation analysis generally illustrates the pairwise associations for a set of variables and decides their direction and strength [30]. In this study, correlation analysis was performed using the MATLAB 2015a, while principal component analysis (PCA) of data was employed using the JMP 14 to illustrate geochemical associations within soil heavy metals. To identify the soil contamination from PTEs across the mining area, the empirical Bayesian kriging spatial interpolation method was applied with ArcGIS10.7 for mapping the spatial distribution of given metals.

### 2.4. Potential Ecological Risk Assessment

The potential ecological risk (PER) method was developed by Hakanson (1980) [31], which can be used to evaluate the potential risks, ecological sensitivity, and toxicity of heavy metal concentrations [32,33]. The equations for calculating the PER indexes are as follows:

$$E_r^i = T_r^i \times C_f^i = T_r^i \times (C_i \div S_i) \tag{1}$$

$$RI = \sum_{i=1}^{n} E_r^i \tag{2}$$

where $C_i$ is the measured concentration of heavy metal $i$; $S_i$ is the background value of the contaminant $i$; $C_f^i$ is the single element pollution factor; $E_r^i$ is individual potential ecological risk index; $T_r^i$ is the biological toxicity factor of an individual element, which are defined as Zn = 1, Cu = 10, Cr = 2, Pb = 5, Hg = 40, As = 10. RI is the comprehensive potential ecological risk index.

### 2.5. Potential Human Health Risk Assessment

Human health risk assessment was applied to estimate the health effects of contaminants, and non-carcinogenic and carcinogenic risk hazards were evaluated using the model obtained by the United States Environmental Protection Agency (USEPA) and Integrated Risk Information System (IRIS) [34,35]. For soil heavy metals, there are three exposure routes: inhalation, ingestion, and dermal contact. The chronic daily intake (CDI) values of soil pollutants posed by three pathways were calculated by the following Equations (3)–(5):

$$\text{CDI}_{\text{ingest}} = \left[ (C_{soil} \times I_{ngR} \times \text{EF} \times \text{ED} \times \text{CF}) \div (BW \times AT) \right] \tag{3}$$

$$CDI_{inhale} = [(C_{soil} \times I_{nhR} \times EF \times ED) \div (BW \times AT \times PEF)] \tag{4}$$

$$CDI_{dermal} = [(C_{soil} \times SA \times EF \times ED \times CF \times AF \times ABS) \div (BW \times AT)] \tag{5}$$

where $C_{soil}$ is the concentration of heavy metal in soil (mg·kg$^{-1}$); $I_{ngR}$ is ingestion rate (mg/day$^{-1}$); EF is the exposure frequency (days/year); ED is the exposure duration (years); BW is the average body weight (kg); AT is the averaging time (days); CF is the conversion factor (kg·mg$^{-1}$); $I_{ngR}$ is inhalation rate (mg/day); AF is the skin adherence factor (mg·cm$^{-3}$); SA is the exposed area of the skin (cm); ABS is the dermal absorption coefficient. Health risk assessment equation variables are given in Table 1.

**Table 1.** Potential risk degree of six heavy metals.

| Potential Ecological Risk (PER) Index | | | | | |
|---|---|---|---|---|---|
| **Class** | $E^i_r$ | **Risk Classification** | **Class** | **RI** | **Risk Classification** |
| A | $E^i_r < 40$ | Low | A | RI < 150 | Low |
| B | $40 \leq E^i_r < 80$ | Moderate | B | $150 \leq$ RI < 300 | Moderate |
| C | $80 \leq E^i_r < 160$ | Considerable | C | $300 \leq$ RI < 600 | High |
| D | $160 \leq E^i_r < 320$ | High | D | RI $\geq$ 600 | Serious |
| E | $E^i_r \geq 320$ | Very high | | | |

Hazard quotient (HQ) indicates the non-carcinogenic risk of an exposure pathway during the exposure duration [36]. Hazard index (HI) is the sum of the HQs, which assesses the total health risk posed by PTEs via exposure pathways [37]. HQ and HI indexes were calculated based on the following Equations (6) and (7):

$$HQ = CDI \div RfD \tag{6}$$

$$HI = \sum HQ = HQ_{ing} + HQ_{inhale} + HQ_{dermal} \tag{7}$$

where RfD (mg/kg/day) is the reference dose values of heavy metals; the RfD values of three exposure pathways are showed in Table 2. HI > 1 suggests that the possible occurrence of non-carcinogenic risk effects exists and HI < 1 is regarded as no significant risk of non-carcinogenic effects from exposures to the heavy metal concentration in studied ingredients [38].

**Table 2.** Parameters applied for assessing the human health risk.

| Parameters | Units | Value | | Reference |
|---|---|---|---|---|
| | | **Adult** | **Children** | |
| $I_{ng}R$ | mg/day | 100 | 200 | [39] |
| EF | day/year | 350 | 350 | [39] |
| ED | year | 26 | 6 | [39] |
| SA | cm$^2$ | 6032 | 2373 | [39] |
| AF$_{soil}$ | mg/cm$^2$ | 0.07 | 0.2 | [39] |
| ABS | / | 0.001 (0.03 for As) | 0.001 (0.03 for As) | [40] |
| PEF | m$^3$/kg | $1.36 \times 10^9$ | $1.36 \times 10^9$ | [39] |
| BW | kg | 15 | 80 | [39] |
| AT$_{non-carcinogenic}$ | days | $365 \times$ ED | $365 \times$ ED | [39] |
| CF | kg/mg | $1 \times 10^{-6}$ | $1 \times 10^{-6}$ | [39] |
| $I_{nhR}$ | m$^3$/days | 20 | 7.6 | [39] |
| AT$_{carcinogenic}$ | days | $365 \times 70$ | $365 \times 70$ | [39] |

A cancer risk (CR) estimates an incremental probability of carcinogenic hazards during a lifetime of seventy years [41]. The lifetime cancer risk (LCR) was computed through cumulating the cancer risk value of each exposure pathway by using Equations (8) and (9).

$$\text{CancerRisk} = CDI_{ing\backslash inhale\backslash dermal} \times CSF \tag{8}$$

$$\text{LCR} = \sum CancerRisk = CancerRisk_{ing} + CancerRisk_{inhale} + CancerRisk_{dermal} \tag{9}$$

where cancer slope factor (CSF) coefficient is used to estimate the risk of cancer connected with exposure to a carcinogenic or potentially carcinogenic ingredients (Table 3). LCR = $1.0 \times 10^{-6}$–$1.0 \times 10^{-4}$ is regarded as an acceptable or inconsequential risk [42].

**Table 3.** Reference doses (RfD) values of non-carcinogenic metals and slope factor (SF) values of carcinogenic metals.

| Heavy Metal | RfD mg/(kg·d) [38] | | | CSF (kg·d)/mg [36] |
|---|---|---|---|---|
| | Ingest | Inhale | Dermal | |
| Zn | 0.300 | 0.300 | 0.060 | / |
| Cu | 0.040 | 0.040 | 0.012 | / |
| Cr | 0.003 | 0.0000286 | 0.00006 | 0.5 |
| Pb | 0.0035 | 0.00352 | 0.000525 | 0.0085 |
| Hg | 0.0003 | 0.0003 | 0.000024 | / |
| As | 0.0003 | 0.000123 | 0.0003 | 1.5 |

## 3. Results and Discussion

### 3.1. Statistical Analysis of Heavy Metal Concentrations

Table 4 shows that the average contents of Zn, Cu, Cr, Pb, Hg, and As were 46.0, 18.92, 99.53, 13.49, 0.06, and 27.94 mg.kg$^{-1}$, respectively. The mean concentrations of Cr, Hg, and As were 3.4, 1.0, and 2.8 times greater than their corresponding provincial Background values had standard rates of 73.33%, 76.67%, and 83.33%, whereas Zn, Cu, and Pb were in the allowed range. Based on the national soil application function and protection target, the national soil environmental quality standard I and standard II [43] (GB15618-1995) are regarded as the threshold value to maintain the natural ecology of the region and limit value to protect the human health [44]. In this study, concentrations of Zn, Cu, Pb, and Hg were significantly lower than both standards. However, Cr contents in soil exceeded the standard I in a minimal size, whereas As contents were relatively higher than standard II. The PH value of samples ranged from 7.16 to 8.86 with a mean value of 8.31, which significantly exceeded the national soil environmental quality standard II (pH > 7.5). The above results depict that the soils in the coal mining area were contaminated by the PTEs in varying degrees, and Cr, Hg and As maybe the dominant pollutants in the area due to the high over standard ratio.

In the descriptive statistics, a coefficient of variation greater than 50% indicates that spatial distributions of PTEs have been affected by significant external interferences [45]. It can be seen from the results of coefficients of variation (CV) of six PTEs that all metals had variation ranging from 16% (Zn)~71% (Hg), with the decreasing order of Hg > As > Cr > Pb > Cu > Zn. The higher CV values of Cr, Hg, and As, which were greater than 50%, indicated a strong influence of external factors. The relatively small CV value (CV < 30%) for Zn, Cu, and Pb illustrated the significant impact of natural factors such as geological properties on the accumulation of these three metals in soil.

**Table 4.** Descriptive statistics of heavy metal concentrations in soil.

|  | Mean | Min | Max | SD | CV% | Background[P] | Over Standard % | Standard I | Standard II |
|---|---|---|---|---|---|---|---|---|---|
| Zn | 46.00 | 25.41 | 58.43 | 7.39 | 16 | 68.80 | / | 100 | 250 |
| Cu | 18.92 | 13.32 | 25.99 | 3.73 | 20 | 26.70 | / | 35 | 100 |
| Cr | 99.53 | 41.65 | 207.29 | 50.83 | 51 | 49.30 | 73.33 | 90 | 250 |
| Pb | 13.49 | 5.48 | 18.49 | 3.92 | 29 | 19.40 | / | 35 | 300 |
| Hg | 0.06 | 0.01 | 0.15 | 0.04 | 71 | 0.02 | 76.67 | 0.15 | 0.5 |
| As | 27.94 | 0.81 | 66.95 | 14.86 | 53 | 11.20 | 83.33 | 15 | 25 |

Background[P]: Soil background values in Changji Prefecture, Xinjiang, China. Standard I: The China national soil environmental quality standard I. Standard II: The China national soil environmental quality standard II [43] (GB15618-1995).

### 3.2. Spatial Distribution Patterns of PTEs

From the spatial distribution pattern of soil PTEs pollution in the study area (Figure 2), it can be seen that the similar spatial distribution of Zn and Cu showed a symmetrical distribution, the decreasing trends between 0–300 m and increased from 300–500 m, and the lowest accumulation levels of Zn and Cu were evenly distributed between 200–350 m. The distribution patterns of these metals exhibited less connectivity with distance and illustrated a non-point source pollution. The average contents of them were well below the provincial background levels, and explained no notable effects of anthropogenic activities.

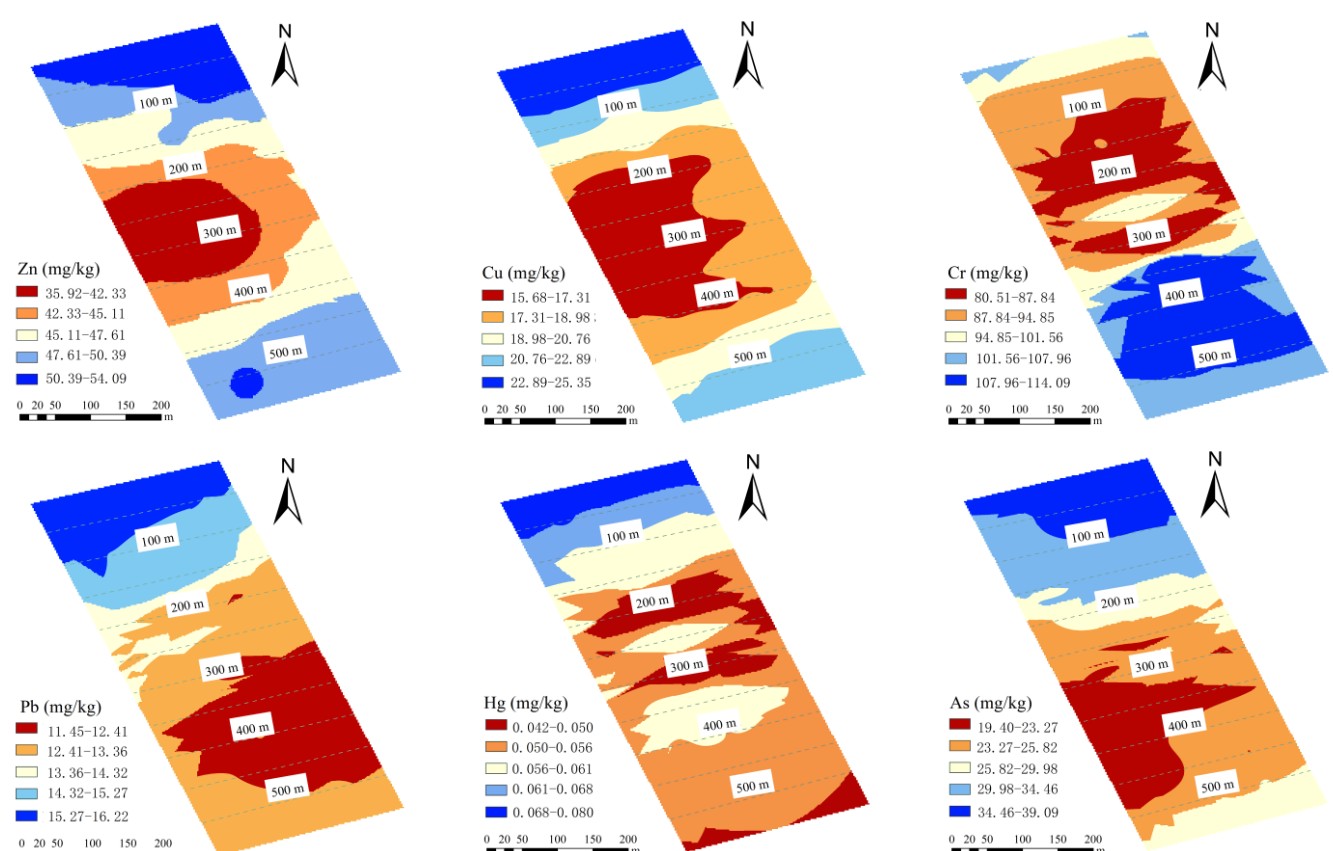

**Figure 2.** Spatial distribution patterns of six PTEs in soils.

There was not any regularity in the spatial distribution of Cr and Hg, as spatial patterns of these elements were not linked with the distance. The irregular distribution patterns explained significant external interferences. High concentration value and coefficient of variations of Cr and Hg that were higher than 50% also confirmed this point.

The concentrations of Pb and As gradually decreased with the increasing of distance away from the open pit coal mine between the ranges of 0–500 m and this trend was in line with the predominant wind direction (NW). In contrast to the As, high concentration values of Pb occurred in the southwest part, while high accumulation values of As were observed in the northeast part of the study area. These results imply the origin of different emission sources connected with the mining activities.

### 3.3. Multivariate Statistical Analysis and Probable Sources of PTEs

Correlation analysis results for six PTEs (Figure 3) showed that the significant positive correlations found between Zn and Cu ($R^2 = 0.73$, $p < 0.01$), as well as between Hg and Cr ($R^2 = 0.74$, $p < 0.01$), indicate that they come from similar pollution sources. The notable positive correlations between the groups Cr-Zn-Cu (0.37, 0.41) and Hg-Cu-Pb (0.39, 0.45) ($p < 0.05$) revealed that these metals may be associated with similar geo-chemical behaviors and common pollution origins. Notable negative correlation existed between As and Cr ($r = -0.36$, $p < 0.05$), whereas weak negative and positive correlations appeared between Pb and other PTEs (except Hg) in all sampling sites.

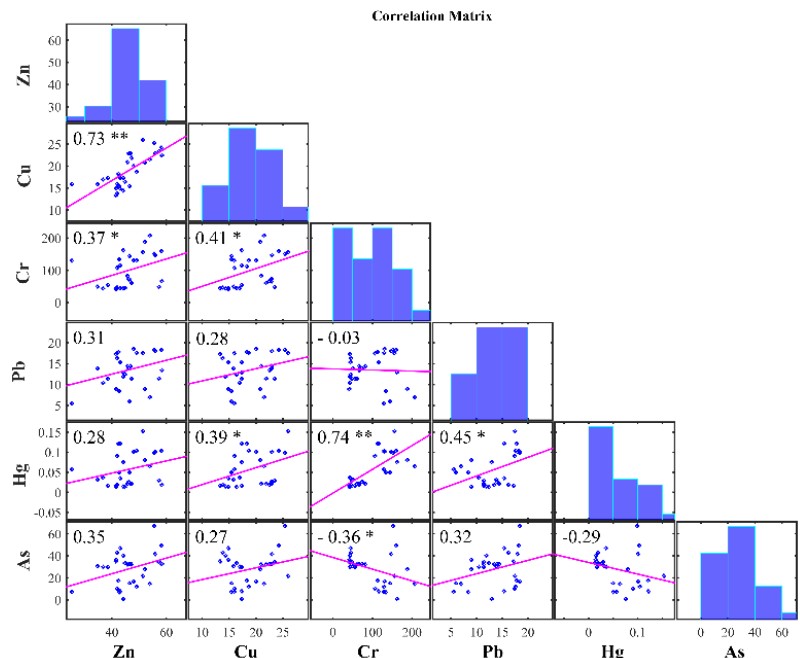

**Figure 3.** Correlation matrix for six PTEs in soil ** $p < 0.01$: Correlation significant at 0.01 level. * $p < 0.05$: Correlation significant at 0.05 level.

The above results indicated that the pollutants in the study area might come from different sources. In order to further identify the pollution sources, principal component analysis was carried out.

In the PCA, principal components with eigenvalue >1 are regarded as significant [46]. In this study, two principal components with initial eigenvalues higher than 1 were extracted, which accounted for 72.7% of the variance based on the Kaiser–Meyer–Olkin (KMO = 0.602) and Bartlett sphericity ($p = 0.000$) tests (Figure 4). The results showed that the first principal component (PC1) contributed 43.6% of the total variance and explained strong positive loads for Zn, Cu, Cr, and Hg. The second principal component (PC2) indicated 29.1% of the total variance and was dominated by a strong positive loading of As. In contrast to As value, Zn and Cu demonstrated strong positive loadings in PC1 and low loadings in PC2. Pb explained moderate positive loadings in both components, whereas significant positive and negative loadings appeared for Cr and Hg in PC1 and

PC2. According to the PCA results and correlations between the six heavy metals, these metals can be categorized into four groups.

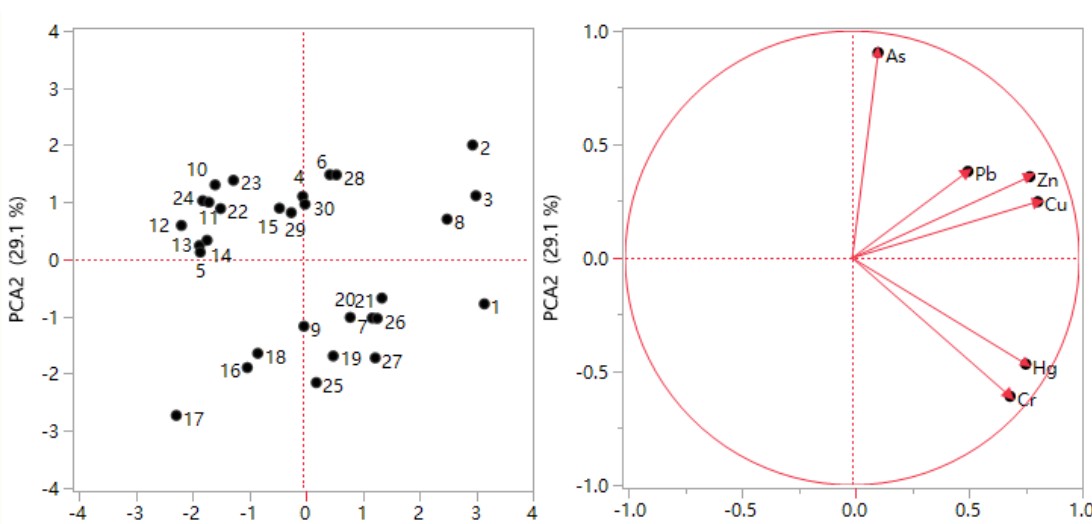

**Figure 4.** Results of principal component analysis (PCA) on six heavy metals.

Group1: Zn-Cu

The results of geostatistical and multivariate statistical analysis of current study have illustrated the identical coming sources of Cu and Zn concentrations in soils of mining area, for similar distribution patterns and strong positive correlation coefficient ($R^2 = 0.74$), together with the significant positive loadings in PC1. Lower concentration values were significantly lower than their corresponding background levels, and the disconnection between the distribution patterns with distance from the coal mine explained the influence of natural factors. The result of this study is consistent with the conclusions of the research works on heavy metal pollution sources and risk assessment in soils of the mining area, and Wucaiwan (Junggar basin) showed that the concentration of Zn and Cu was mainly related to soil parent materials [46–48]. Therefore, the accumulation of Zn and Cu in the soil around the mining area is mainly controlled by natural sources.

Group 2: Hg-Cr

The results of Pearson's correlation relationship between Cr and Hg ($R^2 = 0.74$) suggest similar pollution sources, and this is consistent with the results of PCA grouping. The irregular distribution patterns and high CV values greater than 50% indicate a strong anthropogenic influence. The mean Cr and Hg concentrations all exceeded their corresponding background values by being over the standard rate of 73.33% and 76.67%, respectively, indicating a significant anthropogenic influence. Industrial waste disposal, sewage sludge, spills, and residues all lead to the accumulation of Cr [49]. Studies on the metal source assessment also showed that high contents of Hg in soils in China were mainly caused by industrial activities, including metallurgy, coal-fired, and coal mining industries [50]. As mentioned, there were coal-power plants, new material, mining, and chemical industries around the study area. Therefore, industries activities were the main source of Cr and Hg accumulation in the soils of the study area.

Group3: As

The mean value of As was almost three times greater than the corresponding background value and gradually decreased with the distance away from the coalmine. Higher concentration of As at different distances appeared in the northeast of the study area, close to the coal mine that being mined compared to the corresponding values in the southwest. These results explained that continued mining activities could be the main factor for the

serious accumulation of As. Numerous studies have demonstrated that the average content of As in coal of China was higher than in other countries, and it is estimated that more than 60% of As originated from coal combustion sources [51]. Atmospheric deposition can be transmitted over long distances, which may be a long-term response caused by perennial winds. This result suggested that As originated from coal combustion released during the mining activities.

Group 4: Pb

In this study, the mean value of Pb was well below its background value, indicating the significant impact of natural factors related to geological background. The research results of Huang et al. [52,53] illustrated that Pb originated from natural sources. However, in contrast to As, the high values of Pb occurred in the southwest part of the area (Figure 2), closer to the main roads. Pb is the primary marker of traffic emissions, and roadway and transportation were the major sources of Pb concentration in the mining districts, where trucks, motor vehicles, and equipment will discharge exhaust with Pb, causing pollution in soils [54]. Therefore, transportation has been giving rise to the accumulation of lead content to some degree. Through the above results, it can be seen that Pb comes from mixed sources due to the predominant natural sources and traffic emission.

### 3.4. Assessment of Potential Ecological Risk (PER)

In order to evaluate the external influences on contaminants around the coal mine in the study area, basic statistical analysis was applied to calculate the statistical characteristics of potential ecological risk indexes. From Table 5 it can be seen that the elements of Zn, Cu, Cr, and Pb did not exceed the limits of PER, indicating a low risk. Mean contents with standard deviations of potential ecological risk indexes for six metals decreased in the order of Hg > As > Cu > Cr > Pb > Zn. As a second dominant risk factor, $E_r^i$ values of As were relatively low due to 90% of samples being classified as level A, indicating that only 10% of samples posed moderate risks. The $E_r^i$ values of Hg ranged from 25.05 to 306.68 with a mean value of 113.64, contributing 77% of ecological risks in which 27%, 17%, and 33% of them were in class B, class C, and class D, respectively, implying that Hg was the most serious risk factor. The RI values of multi-metals were in the range of 66.27 to 347.11 with a mean value of 153.47, and 60% of samples under the class A threshold value belong to the lower risk degree, whereas almost two-thirds of samples were in the moderate risk level. Generally, industrial activities were considered as responsible factors for the high ecological impact of Hg.

**Table 5.** Statistical analysis of potential ecological risk of heavy metals.

|     | Average | Min | Max | SD | CV% | Skewness | Kurtosis | Class/Proportion% |
|-----|---------|-----|-----|-----|-----|----------|----------|-------------------|
| Zn | 0.28 | 0.19 | 0.38 | 0.05 | 20 | 0.32 | −1.12 | A/100% |
| Cu | 7.09 | 4.99 | 9.73 | 1.40 | 20 | 0.30 | −1.22 | A/100% |
| Cr | 4.04 | 1.69 | 8.41 | 2.06 | 51 | 0.38 | −1.12 | A/100% |
| Pb | 3.48 | 1.41 | 4.77 | 1.01 | 29 | −0.48 | −0.78 | A/100% |
| Hg | 113.64 | 25.05 | 306.68 | 81.80 | 72 | 0.67 | −0.76 | A/23%, B/27%, C/17%, D/33% |
| As | 24.95 | 0.73 | 59.77 | 13.27 | 53 | 0.26 | 0.24 | A/90%, B/10% |
| RI | 153.47 | 66.27 | 347.11 | 81.62 | 53 | 0.79 | −0.68 | A/60%, B/37%, C/3% |

### 3.5. Potential Human Health Risk

Soil PTEs may cause greatest harm to human health and the eco-environment due to their toxicity and extensive potential impacts. Non-carcinogenic and carcinogenic health risk assessments for both adults and children were estimated and are presented in Tables 6 and 7 and Figure 5. The total HQs of soil PTEs ranged from $6.54 \times 10^{-4}$ to $1.57 \times 10^{-1}$ and from $1.33 \times 10^{-3}$ to 1.67, with a total HI value of $1.81 \times 10^{-1}$ and 1.82, respectively. Regardless of the adults or children, the HI values can be sequenced in the descending order of As > Cr > Pb > Cu > Zn > Hg. As, Cr, and Pb were regarded as the

main elements causing non-carcinogenic risks to children. HIAs for children were almost ten times greater than HIAs for adults and exceeded the safety threshold value 1, whereas HI value for adults was within the acceptable level (HI < 1), indicating that As was the predominant contributor for non-carcinogenic risk through ingestion, followed by Cr and Pb. Similar studies also revealed that the non-carcinogenic risk of children exposed to contaminated environment was higher than that of adults.

**Table 6.** Soil PTEs for non-carcinogenic risk.

|  |  | Zn | Cu | Cr | Pb | Hg | As | Total |
|---|---|---|---|---|---|---|---|---|
| $HQ_{ing}$ | Adults | $1.84 \times 10^{-4}$ | $5.67 \times 10^{-4}$ | $3.98 \times 10^{-2}$ | $4.62 \times 10^{-3}$ | $2.27 \times 10^{-4}$ | $1.12 \times 10^{-1}$ | $1.57 \times 10^{-1}$ |
|  | Children | $1.96 \times 10^{-3}$ | $6.05 \times 10^{-3}$ | $4.24 \times 10^{-1}$ | $4.93 \times 10^{-2}$ | $2.42 \times 10^{-3}$ | 1.19 | 1.67 |
| $HQ_{inh}$ | Adults | $2.70 \times 10^{-8}$ | $8.34 \times 10^{-8}$ | $6.13 \times 10^{-4}$ | $6.76 \times 10^{-7}$ | $3.34 \times 10^{-8}$ | $4.00 \times 10^{-5}$ | $6.54 \times 10^{-4}$ |
|  | Children | $5.48 \times 10^{-8}$ | $1.69 \times 10^{-7}$ | $1.24 \times 10^{-3}$ | $1.37 \times 10^{-6}$ | $6.7 \times 10^{-8}$ | $8.12 \times 10^{-5}$ | $1.33 \times 10^{-3}$ |
| $HQ_{derm}$ | Adults | $3.88 \times 10^{-6}$ | $7.98 \times 10^{-6}$ | $8.40 \times 10^{-3}$ | $1.30 \times 10^{-4}$ | $1.20 \times 10^{-5}$ | $1.41 \times 10^{-2}$ | $2.27 \times 10^{-2}$ |
|  | Children | $2.33 \times 10^{-5}$ | $4.78 \times 10^{-5}$ | $5.03 \times 10^{-2}$ | $7.80 \times 10^{-4}$ | $7.18 \times 10^{-5}$ | $8.48 \times 10^{-2}$ | $1.36 \times 10^{-1}$ |
| HI | Adults | $2.43 \times 10^{-4}$ | $4.89 \times 10^{-2}$ | $4.77 \times 10^{-3}$ | $4.77 \times 10^{-3}$ | $2.39 \times 10^{-4}$ | $1.26 \times 10^{-1}$ | $1.81 \times 10^{-1}$ |
|  | Children | $2.57 \times 10^{-3}$ | $6.34 \times 10^{-3}$ | $4.77 \times 10^{-1}$ | $5.02 \times 10^{-2}$ | $2.49 \times 10^{-3}$ | 1.28 | 1.82 |

**Table 7.** Carcinogenic risk (CR) and lifetime cancer risk (LCR) hazard index for adults and children.

| Metals | Adults | | | | Children | | | |
|---|---|---|---|---|---|---|---|---|
|  | $CR_{ing}$ | $CR_{inh}$ | $CR_{derm}$ | LCR(HI) | $CR_{ing}$ | $CR_{inh}$ | $CR_{derm}$ | LCR(HI) |
| Cr | $7.39 \times 10^{-3}$ | $1.14 \times 10^{-4}$ | $1.56 \times 10^{-3}$ | $9.06 \times 10^{-3}$ | $1.82 \times 10^{-2}$ | $5.33 \times 10^{-5}$ | $2.16 \times 10^{-3}$ | $2.04 \times 10^{-2}$ |
| Pb | $1.46 \times 10^{-5}$ | $2.13 \times 10^{-9}$ | $4.11 \times 10^{-7}$ | $1.50 \times 10^{-5}$ | $3.59 \times 10^{-5}$ | $9.98 \times 10^{-10}$ | $5.68 \times 10^{-7}$ | $3.65 \times 10^{-5}$ |
| As | $6.22 \times 10^{-2}$ | $2.23 \times 10^{-5}$ | $7.88 \times 10^{-3}$ | $7.01 \times 10^{-2}$ | $1.53 \times 10^{-1}$ | $1.04 \times 10^{-5}$ | $1.09 \times 10^{-2}$ | $1.64 \times 10^{-1}$ |
| Total | $6.96 \times 10^{-2}$ | $1.36 \times 10^{-4}$ | $9.44 \times 10^{-3}$ | $7.92 \times 10^{-2}$ | $1.71 \times 10^{-1}$ | $6.37 \times 10^{-5}$ | $1.31 \times 10^{-2}$ | $1.84 \times 10^{-1}$ |

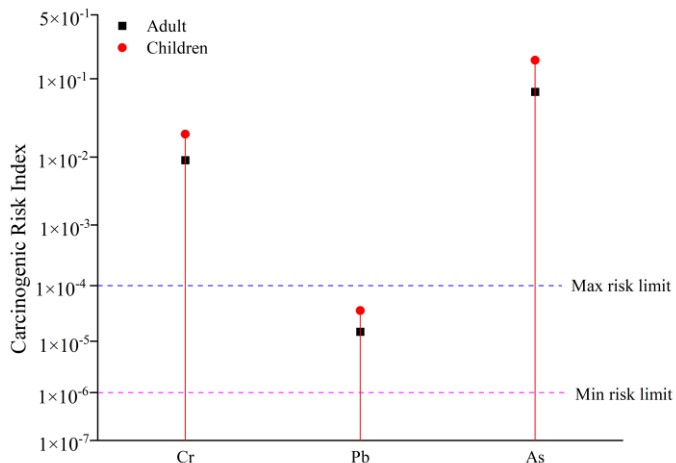

**Figure 5.** Carcinogenic risk for adults and children.

The carcinogenic risk (CR) results of heavy metals (Cr, Pb, and As) are shown in Table 7. It can be seen from the figures and tables that the LCR values of adults and children are $7.01 \times 10^{-2}$ (As) $-1.50 \times 10^{-5}$ (Pb) and $1.64 \times 10^{-1}$ (As) $-3.65 \times 10^{-5}$ (Pb), respectively. Children's lifetime carcinogenic risk ($1.84 \times 10^{-1}$) is more than twice that of adults ($7.92 \times 10^{-2}$), indicating that children are more susceptible to the potential effects of specific metals in soil. The LCR values of all PTEs were higher than the acceptable limit of

$1.0 \times 10^{-6}$ and decreased in the order of As > Cr > Pb. The LCR of As and Cr was the highest, significantly exceeding the maximum risk limit of $1.0 \times 10^{-4}$, indicating that industrial activities such as mining and metal processing may pose a greater threat to human health, and children are more at risk than adults.

Figure 5 clearly shows the carcinogenic risk levels of these three elements for adults and children, from which it can be seen that element Cr and element As exceed the maximum risk limit, and element Pb is below the maximum risk limit, and it can be inferred that the study area as a whole is contaminated with element Cr and element As. These two elements have relatively high levels of carcinogenic risk in soil.

The non-carcinogenic risk level of potential ecological risk elements in the soil of adults and children in the study area is shown in Figure 6. Considering the non-carcinogenic and carcinogenic health risk assessment, As is the primary risk element, followed by Cr, which exceeds the acceptable level. Compared with adults, children tend to have a higher non-carcinogenic risk due to the behavioral and physiological characteristics of hand–mouth activities on soil, indicating that they are more susceptible to soil pollutants. The total HQ and CR values of non-carcinogenic and carcinogenic risk pathways indicated that the risk of PTEs through the ingestion pathway accounted for the majority and followed the order of ingestion > skin > inhalation. This result is consistent with the conclusions of previous studies.

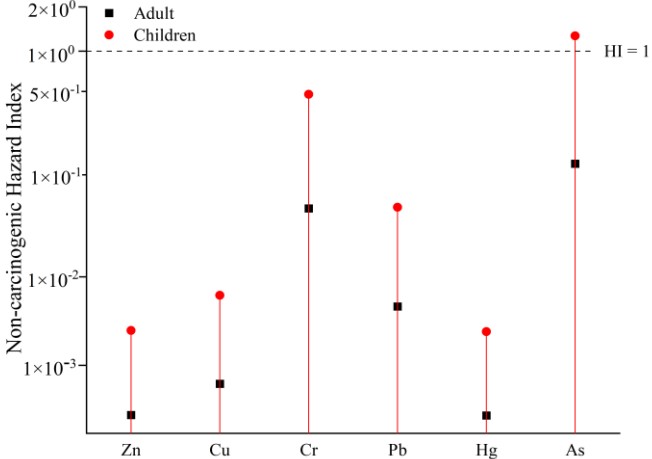

**Figure 6.** Hazard indexes of non-carcinogenic.

## 4. Conclusions

The conclusions of the current study demonstrate that mean concentrations of Cr, Hg, and As were 3.4, 1.0, and 2.8 times greater than corresponding provincial background values, and the average contents of Cr and As exceeded their national soil environmental quality standard value I and II, respectively. However, the concentrations of Zn, Cu, and Pb were lower than background values and were within the allowable level.

Based on the results of geostatistical and multivariate statistical analysis, the current study categorized the six PTEs into four groups. Group 1 (Zn and Cu) was mainly from natural factors related to geological background. Group 2 (Cr and Hg) originated from industrial practices. Group 3 (As) was associated with coal combustion during the mining activities. Group 4 (Pb) explained a mixed source of dominant natural influences and the joint effect of vehicular emission during the mining activities.

Considering both non-carcinogenic and carcinogenic health risk assessment, As was the primary risk element, followed by Cr, due to exceeding the acceptable level. HIAs for children were almost ten times greater than HIAs for adults, and LCR for children was more than two times higher than that for adults, revealing that children have higher health risks than adults. The total HQ and CR values indicate that the risk of PTEs was released during the industrial activities, although the ingestion accounted for most of the risk.

The findings of this study would be informative for understanding the potential ecological and health risks of PTEs in the soils of mining areas in arid regions, suggesting the vitality of controlling the emission and waste material sources during mining and other industrial activities for the prevention of the potential hazards on ecology and human health.

**Author Contributions:** A.A. and B.I. designed and wrote this paper, performed the experiments, and conducted the fieldwork; A.A. conceived this study; H.A. was responsible for manuscript proofreading and data analysis. All authors have read and agreed to the published version of the manuscript.

**Funding:** The Chinese National Natural Science Foundation" Assessment and source identification of heavy metal exposures of wild animals in Xinjiang Kalamaili Mountain Nature Reserve" (No. 42167058).

**Institutional Review Board Statement:** Not applicable.

**Informed Consent Statement:** Not applicable.

**Data Availability Statement:** Not applicable.

**Acknowledgments:** The authors are grateful to the anonymous reviewers for their useful comments and detailed suggestions for this manuscript.

**Conflicts of Interest:** The authors declare no conflict of interest.

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
