# Peer review of "Spatial Patterns, Possible Sources, and Risks Assessment of Soil Potentially Toxic Elements in an Open Pit Coal Mining Area in a Typical Arid Region"

_sustainability, doi:10.3390/su151612432_

Round 1

Reviewer 1 Report

Dear Editor:

Thank you for giving me the opportunity to revise the MS entitled “Spatial patterns, possible sources and potential risks of soil heavy metals in an open pit coal mining area in a typical arid region” by Abdugheni Abliz and his/her colleagues that was submitted to “sustainability”. The MS submitted is suitable for sustainability, and some interesting results were showed. However, there are several requirements that have to consider by the authors. In this regard, the following comments are requested to be addressed by the authors: 

1.       The Abstract section needs to be carefully revised. The innovation of the manuscript must be clearly stated in the text.

2.       The citation form in the text should be unified. For example, Line 175 “(Zhang et al.,2018), (Kusin et al.,2018)”.

3.       Please carefully check the Figure and table and make the layout of the Figures more aesthetically pleasing, such as Figure 5, Table 6.

4.       The discussion is not in-depth enough, please strengthen the discussion and comparative analysis

5.       Please note that the format of references must be uniform. For example, Line 534. In addition, some references are outdated.

I would suggest that the authors review and include the following recent studies to improve the manuscript.

Organic–inorganic composite modifiers enhance restoration potential of Nerium oleander L. to lead–zinc tailing: application of phytoremediation. Environ Sci Pollut R. 2023, DOI:10.1007/s11356-023-26359-w. 

Best regards,

Author Response

First of all, we would like to thank the reviewers for the constructive comments and suggestions our paper “Spatial patterns, possible sources and potential risks of soil heavy metals in an open pit coal mining area in a typical arid region”, they were very valuable and useful. We have now revised our manuscript according to the valuable comments and suggestions addressed by the reviewers and incorporated these into revisions. The blue color represents the revised parts in manuscript. The detailed response to reviewers is listed below:

Reviewer 1

Dear Editor:

Thank you for giving me the opportunity to revise the MS entitled “Spatial patterns, possible sources and potential risks of soil heavy metals in an open pit coal mining area in a typical arid region” by Abdugheni Abliz and his/her colleagues that was submitted to “sustainability”. The MS submitted is suitable for sustainability, and some interesting results were showed. However, there are several requirements that have to consider by the authors. In this regard, the following comments are requested to be addressed by the authors:

 Author response: Thank you very much to the reviewers for their valuable time in reviewing the manuscripts and for their positive and valuable comments.

  1. The Abstract section needs to be carefully revised. The innovation of the manuscript must be clearly stated in the text.

Author response: We are very grateful to the reviewers for pointing out this issue. It has been revised depending on the reviewer comments. At the same time, the overall revision of the paper was made, and the clarity and fluency of the textual presentation was further improved, hence the change in line numbering. (Please see the abstract part in revised manuscript)

  1. The citation form in the text should be unified. For example, Line 175 “(Zhang et al.,2018), (Kusin et al.,2018)”.

Author response: Thank you very much to the reviewers for pointing out this problem, it was corrected. Due to the corresponding improvements in the abstract and introduction of this article, the line number has changed. (Please see the revised version line175 and full-text)

  1. Please carefully check the Figure and table and make the layout of the Figures more aesthetically pleasing, such as Figure 5, Table 6.

Author response: Thanks for reviewer pointing out the shortcoming, we are extremely grateful to the reviewer for pointing out this mistake. According to the reviewer 's point of view, it was corrected. (Please see the revised version of the manuscript)

  1. The discussion is not in-depth enough, please strengthen the discussion and comparative analysis.

Author response: Many thanks to the reviewers for pointing out this problem. we carefully read the submission notes of your journals and revised the paper as a whole according to the opinions of the reviewers, it was corrected. (Please see the revised version of the manuscript)

  1. Please note that the format of references must be uniform. For example, Line 534. In addition, some references are outdated.

Author response: Thanks for the valuable comments. We have carefully modified the relevant issues. (Please see the revised version of the manuscript)

  1. I would suggest that the authors review and include the following recent studies to improve the manuscript. Organic–inorganic composite modifiers enhance restoration potential of Nerium oleander L. to lead–zinc tailing: application of phytoremediation. Environ Sci Pollut R. 2023, DOI:10.1007/s11356-023-26359-w.

Author response: Many thanks to the reviewers for pointing out this shortcoming. It was corrected. (Please see the revised version line64-104 and line405-456.)

The newly added references are:

[1] Su, R. K.; Ou, Q. Q.; Wang, H. Q.; Dai, X. R.; Chen, Y. H.; Luo, Y. T.; Yao, H. S. Ou Yang, D. X.; Li, Z. S.; Wang, Z. X.  Organic–inorganic composite modifiers enhance restoration potential of Nerium oleander L. to lead–zinc tailing: application of phytoremediation[J]. Environ Sci Pollut Res, 2023, 30(19): 56569-56579.https:// doi:10.1007/s11356-023-26359-w

[25] YUANG Ting-ting, WANG Zhi-qiang, WANG Xi-yuan, MA Yu, HAN Yong, ZHANG Zi-guang. Buffer Analysis of Heavy Metal Ecological Risk in the Hongshaquan Mining Area of East Junggar Basin. Chinese J Soil Sci, 2020, 51(1): 227-233. http://doi.org/10.19336/j.cnki.trtb.2020.01.31

[3] Bai, J. F.; Zhang, S. J.; Gu, W. H.; Gu, D.; Dong, B.; Zhao, J.; Hu, J.; Chen, J. M. Chapter 13-Bioleaching of heavy metals from a contaminated soil using bacteria from wastewater sludge. Sustainable and Circular Management of Resources and Waste Towards a Green Deal. 2023, 183-198. https://doi.org/10.1016/B978-0-323-95278-1.00018-8

Thanks again to the reviewers for the time and energy they have spent on our manuscripts. We carefully read the comments and made corrections, hoping to get everyone 's approval. If there are still shortcomings, please point out that we will try our best to change it to your satisfaction.

Sincerely,

Author: Abdugheni Abliz

Email: [email protected]/[email protected]

Reviewer 2 Report

The paper titled “Spatial patterns, possible sources and potential risks of soil heavy metals in an open pit coal mining area in a typical arid region” deals with the investigation of six metals concentrations in the surrounding areas of a coal mine in the tertiary of Hongshaquan mining area (northern Xinjiang, China).

In general, the manuscript is well structured. The subject of the present research is very interesting. The literature is well enriched. The authors are mentioning in their introduction the aims of the present study. However, it would very auspicable if at the end of such an informative introduction, the authors provided clearly the novelty aspects of the present research. Additionally, please refer if any similar studies have been previously carried out in the wider study area. If so, then please state why this study excels compared to other similar. In my opinion, this would make the manuscript more interesting to the reader.

Even by the introduction the authors start using the term “heavy metals”. However, as ascertained in the international literature, this term is gradually forsaken. Hence, maybe the authors could consider replacing it by the widely known “potentially toxic elements – PTEs” since in order to become toxic, the concentration of an element must exceed a specified limit, or just by “trace elements” or “metals”. This suggestion should be taken into account for the whole manuscript.

Following that, in the title the authors refer to “HMs Pollution” while in the Statistical and Geo-statistical analysis (part “2.3”), they mention “soil contamination from heavy metals” (in Row 131). Maybe it would be wise to decide whether they are studying contamination or pollution. Because in literature there is specific definition for these two terms. Especially when it comes to PTEs. So, please consider and revise.

Since the manuscript deals with concentrations of elements in soils, it is obvious that the geological setting of the various areas that the samples originate is important. Hence, the authors should consider providing information regarding it, apart from the general information they include in Section 2.1. In addition to that, maybe they could think the possibility of also including geological map of the studied areas, apart from the geographical maps in Figure 1. This would make the manuscript more informative.

The whole investigation of the present paper is based on metals concentrations. However, in Section “2.2 Sample collection and chemical analysis”, the authors just mention how the samples were collected and the method by which each concentration was determined. They do not include the most important information, which is how metals were extracted by the samples. For example, the extraction was performed via the aqua regia digestion or via the 4-acids digestion e.t.c.? Please consider and clarify.

Finally, in some cases there is a difficulty concerning the figures. Their quality should be enhanced. In some cases as in Figures 2 and 5 maybe the diagrams could be separated in order to be enlarged as separated images. That way they will be more comprehensive. Especially in the case of Fig. 3, it definitely needs to be enlarged because it is faded optically.

Regarding the English language style, although I do not feel qualified to judge about it, it must be said that it is very comprehensive and minor spell check is required. For these reasons, I suggest that major revision should be carried out before publication,

Author Response

Reviewer 2

  1. The paper titled “Spatial patterns, possible sources and potential risks of soil heavy metals in an open pit coal mining area in a typical arid region” deals with the investigation of six metals concentrations in the surrounding areas of a coal mine in the tertiary of Hongshaquan mining area (northern Xinjiang, China).

 Author response: Thank you very much for reviewer positive and valuable comments on our manuscript。

  1. In general, the manuscript is well structured. The subject of the present research is very interesting. The literature is well enriched. The authors are mentioning in their introduction the aims of the present study. However, it would very auspicable if at the end of such an informative introduction, the authors provided clearly the novelty aspects of the present research. Additionally, please refer if any similar studies have been previously carried out in the wider study area. If so, then please state why this study excels compared to other similar. In my opinion, this would make the manuscript more interesting to the reader.

 Author response: Many thanks to the reviewer for pointing this out. It was a shortcoming in my writing. I carefully read the previous relevant excellent paper pitches in your journal and revised the paper as a whole based on the reviewer's comments and it was corrected. (Please see the revised version of the manuscript)

  1. Even by the introduction the authors start using the term “heavy metals”. However, as ascertained in the international literature, this term is gradually forsaken. Hence, maybe the authors could consider replacing it by the widely known “potentially toxic elements – PTEs” since in order to become toxic, the concentration of an element must exceed a specified limit, or just by “trace elements” or “metals”. This suggestion should be taken into account for the whole manuscript.

 Author response: Many thanks to the reviewers for their valuable comments, which play a very important role in improving the quality of the manuscript. After discussion among the three authors and review of previous studies, “soil heavy metal elements” were replaced with “soil potentially toxic elements-PTEs” (Please see the revised version of the manuscript)

  1. Following that, in the title the authors refer to “HMs Pollution” while in the Statistical and Geo-statistical analysis (part “2.3”), they mention “soil contamination from heavy metals” (in Row 131). Maybe it would be wise to decide whether they are studying contamination or pollution. Because in literature there is specific definition for these two terms. Especially when it comes to PTEs. So, please consider and revise.

 Author response: Thank the reviewers for their valuable opinions and suggestions. All of our authors decided that the soil samples in the study area were potential risk elements by discussing and consulting relevant studies. Therefore, the soil elements in the study area of the full text were changed to potential ecological risk elements. (Please see the revised version of the manuscript)

  1. Since the manuscript deals with concentrations of elements in soils, it is obvious that the geological setting of the various areas that the samples originate is important. Hence, the authors should consider providing information regarding it, apart from the general information they include in Section 2.1. In addition to that, maybe they could think the possibility of also including geological map of the studied areas, apart from the geographical maps in Figure 1. This would make the manuscript more informative.

Author response: The valuable comments and suggestions of the reviewers are greatly appreciated. This is an important point that the authors overlooked, after reviewing relevant studies. We have added the relevant content. (Please see the revised version of the manuscript)

  1. The whole investigation of the present paper is based on metals concentrations. However, in Section “2.2 Sample collection and chemical analysis”, the authors just mention how the samples were collected and the method by which each concentration was determined. They do not include the most important information, which is how metals were extracted by the samples. For example, the extraction was performed via the aqua regia digestion or via the 4-acids digestion e.t.c.? Please consider and clarify.

 Author response: Many thanks to the reviewers for their valuable comments and suggestions.

has been modified to look like the following:

For identifying the impacts of PTEs toward soils around the open pit coal mine, three sampling lines were selected according to perennial wind direction, and 30 soil samples ground with a wooden stick, and from the 0–20 cm layer was taken with the horizontal average distance interval of approximately 50 m. In order to pinpoint the exact positions of all sampling points, the global positioning system (GPS) was applied to record the exact coordinates. All soil samples were air-dried at room temperature (20–24℃) and sifted through a 2mm nylon mesh sieve to remove stones and other de-bris. About 0.5g soil samples were weighed and placed in polyethylene bags, and ana-lyzing the heavy metal contents of Zn, Cu, Pb, Cr, Hg and As. The Hg and As values were analyzed by the dual-channel automatic atomic fluorescence spectrometer (PF6-2) and the detection limits of these metals were lower than 0.001 and 0.01μg·kg−1 with the degree of precision ≤ 1.0%. In this study, the Hitachi Z-2000 atomic absorption Spectrophotometer was used to determine the Zn, Cu, Cr and Pb concentrations under the detection limits of 0.001, 0.01, 0.004 and 0.002 mg.kg-1. The quality control (QC) was completed using GSS (GSS-8) standard soil sample, provided by the national research center for geo-analysis of China, and the recoveries were within the allowable range for all the metals in the soil [25]. Samples were first digested by the triacid dissolution method (HNO3-HF-HClO4), then determined for As, Zn, Cr, Cu and Pb using a SOL-LARAAM6 atomic absorption spectrometer from Thermo Fisher Scientific, USA. To ensure data accuracy, experimental measurements were quality-controlled with 20% parallel samples, blanks and GSS-1, a standard for soil composition analysis, with a 5% margin of error. All vessels used in the experiments were soaked in a 10% HNO3 solution for over 24 hours. Experimental reagents were of the highest purity, and the water used for the experiments was deionized water.

  1. Finally, in some cases there is a difficulty concerning the figures. Their quality should be enhanced. In some cases, as in Figures 2 and 5 maybe the diagrams could be separated in order to be enlarged as separated images. That way they will be more comprehensive. Especially in the case of Fig. 3, it definitely needs to be enlarged because it is faded optically.

 Author response: Thank you very much for reviewer valuable suggestions. We really appreciate your efforts in reviewing our manuscript. We improved the resolution of Figure 2, Figure 3 and Figure 5 in the manuscript from 200dpi to 500dpi, and the entire manuscript has been screened to prevent possible similar mistake. The revised content is shown below. (Please see the revised version of the manuscript)

  1. Regarding the English language style, although I do not feel qualified to judge about it, it must be said that it is very comprehensive and minor spell check is required. For these reasons, I suggest that major revision should be carried out before publication,

 Author response: We are very grateful to the reviewers for pointing out that we have consulted a large number of documents, a comprehensive understanding of the special geographical and cultural attributes of the study area to write this manuscript. Although we have repeatedly revised and improved, and commissioned a professional polishing agency to do two polishings, there are still some problems in the expression of the manuscript. Thanks again to the reviewer, we checked the manuscript repeatedly according to the requirements, and did a professional polishing agency to do a polishing. (Please see the revised version of the manuscript)

Thanks again to the reviewers for the time and energy they have spent on our manuscripts. We carefully read the comments and made corrections, hoping to get everyone 's approval. If there are still shortcomings, please point out that we will try our best to change it to your satisfaction.

Sincerely,

Author: Abdugheni Abliz

Email: [email protected]/[email protected]
